# Whole Blood Expression Levels of Long Noncoding RNAs: HOTAIRM1, GAS5, MZF1-AS1, and OIP5-AS1 as Biomarkers in Adolescents with Obesity-Related Asthma

**DOI:** 10.3390/ijms24076481

**Published:** 2023-03-30

**Authors:** José J. Leija-Martínez, Carlos A. Guzmán-Martín, Javier González-Ramírez, Abraham Giacoman-Martínez, Blanca E. Del-Río-Navarro, Rodrigo Romero-Nava, Santiago Villafaña, José Luis Flores-Saenz, Fausto Sánchez-Muñoz, Fengyang Huang

**Affiliations:** 1Laboratorio de Investigación en Obesidad y Asma, Hospital Infantil de México Federico Gómez, Ciudad de México 06720, Mexico; leijamjj@gmail.com (J.J.L.-M.);; 2Departamento de Inmunología, Instituto Nacional de Cardiología Ignacio Chávez, Ciudad de México 14080, Mexico; gmcarlos93@gmail.com; 3Laboratorio de Biología Celular, Facultad de Enfermería, Universidad Autónoma de Baja California Campus Mexicali, Mexicali 21280, Mexico; 4Laboratorio de Farmacología, Departamento de Ciencias de la Salud, DCBS, Universidad Autónoma Metropolitana-Iztapalapa (UAM-I), Ciudad de México 09340, Mexico; luisflosaenz@gmail.com; 5Departamento de Inmunología Clínica de Alergia Pediátrica, Hospital Infantil de México Federico Gómez, Ciudad de México 06720, Mexico; 6Laboratorio de Señalización Intracelular, Sección de Estudios de Posgrado e Investigación, Escuela Superior de Medicina, Instituto Politécnico Nacional, Ciudad de México 07738, Mexico

**Keywords:** asthma, obesity-related asthma, long noncoding RNAs, OIP5-AS1, HOTAIRM1, MZF1-AS1, lncRNA GAS5

## Abstract

Asthma is a heterogeneous entity encompassing distinct endotypes and varying phenotypes, characterized by common clinical manifestations, such as shortness of breath, wheezing, and variable airflow obstruction. Two major asthma endotypes based on molecular patterns are described: type 2 endotype (allergic-asthma) and T2 low endotype (obesity-related asthma). Long noncoding RNAs (lncRNAs) are transcripts of more than 200 nucleotides in length, currently involved in many diverse biological functions, such as chromatin remodeling, gene transcription, protein transport, and microRNA processing. Despite the efforts to accurately classify and discriminate all the asthma endotypes and phenotypes, if long noncoding RNAs could play a role as biomarkers in allergic asthmatic and adolescent obesity-related asthma, adolescents remain unknown. To compare expression levels of lncRNAs: HOTAIRM1, OIP5-AS1, MZF1-AS1, and GAS5 from whole blood of Healthy Adolescents (HA), Obese adolescents (O), allergic asthmatic adolescents (AA) and Obesity-related asthma adolescents (OA). We measured and compared expression levels from the whole blood of the groups mentioned above through RT-q-PCR. We found differentially expressed levels of these lncRNAs between the groups of interest. In addition, we found a discriminative value of previously mentioned lncRNAs between studied groups. Finally, we generated an interaction network through bioinformatics. Expression levels of OIP5-AS1, MZF1-AS1, HOTAIRM1, and GAS5 in whole blood from the healthy adolescent population, obese adolescents, allergic asthma adolescents, and obesity-related asthma adolescents are differently expressed. Moreover, these lncRNAs could act as molecular biomarkers that help to discriminate between all studied groups, probably through molecular mechanisms with several genes and miRNAs implicated.

## 1. Introduction

Asthma is a heterogeneous entity, now considered as an “umbrella diagnosis” because it encompasses several distinct diseases or endotypes and varying phenotypes (i.e., obese, young atopic, middle-aged, and elderly), is characterized by common clinical manifestations, such as shortness of breath, wheezing and variable airflow obstruction [1]. This heterogeneity has given rise to a great variety of therapeutic responses, which makes it difficult for clinicians to choose the appropriate treatment for this type of patient [2]. Several authors, including our research team, have contributed to the asthma phenotype and endotype classification, whereby it is currently possible to describe two significant asthma endotypes based on molecular patterns: T2 endotype and T2-low endotype [3,4,5]. T2 endotype or “allergic asthmatic” phenotype is characterized by early onset (before 12 years of age), lymphocytes Th2-dependent allergic inflammation, and eosinophilic bronchial infiltration with elevated serum concentrations of IL-4, IL-5, IL-9, IL-13, and immunoglobulin E [6]. The T2-low endotype, also known as the “non-allergic phenotype” or “obesity-related asthma”, appears after the age of 12 years [7].

In most cases is characterized by the infiltration of neutrophils in the epithelium of the respiratory tract and the appearance of more severe clinical scenarios where TNF-α, IL-1β, IL-6, IL-15, IL-23, IL-17, IL-21, and IL-22 are the centrally implicated cytokines [8].

In this sense, obesity has become a growing pediatric health problem worldwide, affecting more than 124 million children and adolescents worldwide. In addition, and due to several studies that have linked poorer clinical outcomes to “low-grade inflammation”, a key component of obesity, asthma-related obesity has attracted increasing attention from researchers worldwide [9,10].

Nevertheless, even though several authors have focused on accurately classifying and discriminating all the asthma endotypes and phenotypes, there is still a significant knowledge gap between those entities, and certainly, the discovery of new molecular biomarkers is needed [11]. In the search for new disease biomarkers, genomics and transcriptomics studies, in combination with the increased accessibility of new molecular biology technologies, have accelerated the study of noncoding RNAs (ncRNAs) [12]. ncRNAs can be classified based on their length in small and long noncoding RNAs [13]. In this line of information, long noncoding RNAs (lncRNAs) are transcripts of more than 200 nucleotides in length, currently involved in many diverse biological functions, such as chromatin remodeling, gene transcription, protein transport, microRNA processing, and its action has been implicated in several pathologic processes [14,15]. Our group recently showed that whole blood is a suitable sample for characterizing RNA molecules as novel biomarkers in these scenarios [4]. Despite several emerging information have described the potential biological role of lncRNAs in diverse diseases, how lncRNAs could allow discrimination between the two major asthma endotypes remain unknown, and due to the reported links to inflammation, obesity, and lung diseases, we selected the following four lncRNAs HOTAIRM1 [16], GAS5 [17], MZF1-AS1 [18] and OIP5-AS1 [19] as potential biomarkers implicated in asthma. Therefore, our objective was to compare expression levels of these lncRNAs from whole blood of Healthy Adolescents (HA), Obese adolescents (O), allergic asthmatic adolescents (AA), and Obesity-related asthma adolescents (OA). Finally, to characterize our findings, we conducted a bioinformatic analysis through a bioinformatics tool.

## 2. Results

In this study, we analyzed a total of 92 adolescent participants that were divided into four groups: 23 were healthy adolescents, 21 had obesity without asthma, 28 had allergic asthma without obesity, and 20 had obesity-related asthma. The four groups had similar ages, sex distribution, and Tanner distribution. In contrast, anthropometrics significantly differed between Obese and non-obese groups with similar time with obesity. In addition, OA patients had slightly higher Asthma severity with more moderate persistent patients. Finally, treatment was not different between AA and OA. All the main characteristics are presented in Table 1.

HOTAIRM1 and GAS5 showed similar expression behavior, higher expression levels were observed in both asthmatic phenotypes, allergic asthma, and obesity-related asthma compared with their counterparts, healthy and obese adolescents, respectively. Moreover, these lncRNAs do not show statistical differences between healthy donors versus obesity and allergic asthma versus obesity-related asthma: HOTAIRM1 (HA vs. AA, *p* < 0.0001; HA vs. OA, *p* < 0.0001; O vs. AA, *p* < 0.001, and O vs. OA, *p* < 0.001; HA vs. O, *p* > 0.999; AA vs. OA; *p* > 0.999). GAS5 (HA vs. AA, *p* < 0.0001; HA vs. OA, *p* < 0.0001; O vs. AA, *p* = 0.0004, and O vs. OA, *p* = 0.0015; HA vs. O, *p* > 0.999; AA vs. OA; *p* > 0.999), Figure 1a,b.

On the other hand, MZF1-AS1 only showed differences between the Allergic Asthmatic group compared to the others (AA vs. OA, *p* = 0.020, AA vs. O, *p* < 0.001, AA vs. HA, *p* < 0.001) Figure 1c, and OIP5-AS1 expression showed interesting behavior, it was statistically different between all the studied groups, and lower levels were observed in the groups with obesity compared with their counterparts that not presents that comorbidity (HA vs. AA, *p* < 0.001; HA vs. OA, *p* < 0.0001; O vs. AA, *p* < 0.001, and O vs. OA, *p* < 0.001; HA vs. O, *p* < 0.001; AA vs. OA; *p* = 0.024), Figure 1d.

Based on observed differences, we tested the potential discriminative role of the studied lncRNAs. Therefore we performed the area under the curve ROC analysis. In this analysis, we found that HOTAIRM1 and GAS5 may accurately discriminate asthmatic patients from those who have not presented that illness (AUC: 1.00, 95% CI: 1.00–1.00, *p* < 0.0001, and AUC: 0.866, 95% CI: 0.7828 to 0.9492, *p* < 0.0001), respectively Figure 2a. On the other hand, OIP5-AS1 and GAS5 may discriminate between patients with obesity-related asthma from those who only had obesity (AUC:0.9833, 95% CI: 0.9517 to 1.00, *p* < 0.0001, and AUC: 0.8024, 95% CI: 0.6539 to 0.9509, *p* = 0.0009), respectively, Figure 2b. Moreover, OIP5-AS1 and MZF1-AS1 discriminate between the two significant asthma endotypes, AA and OA. Therefore, we obtained the area under the curve ROC of these lncRNAs: OIP5-AS1 (AUC: 0.6938, 95% CI: 0.5434 to 0.8441, *p* = 0.0203) and MZF1-AS1 (AUC: 0.6982, 95% CI: 0.5456 to 0.8509, *p* = 0.0233), respectively, Figure 2c.

Then, due to the high discriminative value of OIP5-AS1 and GAS5, we tested their cumulative value in the abovementioned groups. Therefore, we obtained the expression levels ratio from these lncRNAs based on this new quantitative variable. Next, we carried out an area under the ROC curve analysis, and interestingly this ratio accurately discriminates obesity alone from obesity-related asthma: AUC: 0.9026, 95% CI: 0.7855 to 1.000, *p* < 0001. In this line of analysis, we obtained the optimal cut-off point for the ratio through Youden’s index test; cut-off point = 0.4189, Figure 3a. At the same time, we compared the medians of the OIP5/GAS5 ratio between obesity alone and Obesity-related asthma groups, and in line with previous findings, lower medians of OIP5/GAS5 ratio were observed in the obesity group, *p* < 0.0001, Figure 3b. Finally, using the mentioned cut-off point, we divided all the patients into two groups: those with a ratio above and those below 0.4189. With this new binominal variable, we performed the crosstab Fisher’s Test. We obtained the Odd’s Ratio from the analysis as mentioned above, interestingly those patients with higher OIP5/GAS5 ratio had an OR of 54.0 to belong obesity-related asthma group (95% CI: 8.03 to 362.7, *p* < 0.0001) Figure 3c.

Finally, we generated an interaction network in which we searched the base-pairing interactions of studied lncRNAs with several selected miRNAs and Genes. Interestingly, we found that OIP5-AS1, MZF1-AS1, HOTAIRM1, and GAS5 may indirectly interact with genes described as critical components of the pathophysiology of both significant asthma endotypes, Figure 4.

## 3. Discussion

In this study, we analyzed the expression levels of the lncRNAs: HOTAIRM1, GAS5, OIP5-AS1, and MZF1-AS1 from whole blood of healthy, obese, allergic asthmatic and obesity-related asthmatic adolescents through RT-qPCR. We found differentially expressed levels of these lncRNAs between the groups of interest. In addition, a discriminative value of the abovementioned lncRNAs between diverse studied groups was observed. For instance, HOTAIRM1 and GAS5 may help to discriminate all asthmatic patients from controls. Furthermore, OIP5-AS1 and GAS5 could help to differentiate obesity alone from obesity-related asthma, and OIP5-AS1 and MZF1-AS1 may help to discriminate between asthma endotypes. Finally, we generated an interaction network of the previously mentioned lncRNAs in which several classical genes related to asthma are indirectly involved through the interaction with diverse miRNAs.

First, we found higher levels of lncRNAs: HOTAIRM1, GAS5, MZF1-AS1, and OIP5-AS1 in the allergic asthmatic and obesity-related asthma patients compared with their counterparts’ healthy adolescents and obesity without asthma adolescents, respectively. In this sense, several studies have related higher expression levels of the lncRNAs with more severe pathologic states. For instance, Lipin Wu and collaborators found that higher levels of OIP5-AS1 indicated poor prognosis in pancreatic ductal carcinoma. Meanwhile, the downregulation of the abovementioned lncRNA interfered with cell proliferation and migration, to which miR-429 was related [20]. Furthermore, another study led by Maolong Wang found that OIP5-AS1 was highly expressed in lung cancer tissues and positively correlated with tumor size and growth speed [19].

On the other hand, Xiaoping Wang and colleagues proposed that lncRNA GAS5 may be a key component in the pathophysiology of childhood pneumonia by inhibiting cell apoptosis and promoting SHIP-1 expression via downregulating miR-155 [21]. Furthermore, another study on patients with chronic obstructive pulmonary disease by Rubing Mo and collaborators found that LPS induces lncRNA GAS5 expression and release of IL-2, IL-6, IL-10, and TNF-α. Additionally, the upregulation of GAS5 promoted cell death and inhibited proliferation in the MRC-5 cell line, suggesting that lncRNA GAS5 is related to promoting pyroptosis in that kind of patient [22]. In this line of information, lncRNA HOTAIRM1 has been described to play a critical role in inflammatory and allergic response; one study led by Lihua Li and collaborators found that HOTAIRM1 are implicated in allergic rhinitis by promoting differentiation of T helper type 9 cells through miR-148a-3p interaction [23]. Similarly, in recent years, MZF1-AS1 has been proposed as a promising actor in different pathological phenotypes, including autoimmunity, cancer, and chronic pulmonary diseases [18,24,25]. In sum, these studies suggest that overexpression of OIP5-AS1, GAS5, HOTAIRM1, and MZF1-AS1 play crucial roles in the pathophysiology of several diseases in which more severe states are related; and as we can observe in the bioinformatic interaction pathways image, OIP5-AS1, MZF1-AS1, HOTAIRM1, and GAS5 may indirectly interact with genes and microRNAs that have been described as critical components of the role of low-grade inflammation (i.e., meta inflammation) in some endotypes of asthma, which make us hypothesize that whole blood overexpression of these lncRNAs may be part of a characteristic phenotype of asthmatic patients. In the same way, OIP5-AS1, MZF1-AS1, HOTAIRM1, and GAS5 may be promising molecular targets for developing new therapies for asthmatic patients.

After that, we tested the discriminative value of these lncRNAs through the area under the curve ROC analysis and interestingly found that HOTAIRM1 and GAS5 can help discriminate between any asthma phenotype of healthy and obese patients. Furthermore, the expression levels of OIP5-AS1 and GAS5 may help to discriminate between obesity alone and obesity-related asthma. OIP5-AS1 has little or non-studies in obesity, whereas GAS5 has been implicated in obesity and insulin resistance by targeting some miRNAs, such as miR-21, that have been implicated in obesity, as well as in lung diseases [26]. On the other hand, OIP5-AS1 and MZF1-AS1 appear to accurately differentiate between obese patients from their counterparts presenting with obesity-related asthma. Since several authors have described the potential biomarker role of lncRNAs, various studies have focused on evaluating this study field [27,28]. For instance, Chuling Li and collaborators, through ROC curve analysis, found that lncRNA GAS5 from exosomes combined with carcinoembryonic antigen could be used to distinguish patients with stage I from other stages of non-small cell lung cancer patients with an AUC of 0.822 [29]. Another study led by Qian Tan and collaborators analyzed plasma expression levels of GAS5 from patients with non-small cell lung cancer, and they proposed a combination of GAS5 and CA199 to discriminate early stages of that illness with an area under the curve of 0.734 [30]. In Addition, Feng Xiong and colleagues studied expression levels of HOTAIRM1 on different tumor cell lines of non-small cell lung cancer. They found higher levels of HOTAIRM1 on tumor cell lines compared with controls.

Moreover, they observed that patients with high expression levels of that lncRNA had lower overall survival compared to those with higher expression levels. Finally, the authors suggest that HOTAIRM1 may be a potential diagnostic and prognostic biomarker for such patients [31]. Despite the wide range of information in the field of lncRNAs, studies exploring their potential discriminatory role in different disease states via ROC curve analysis are still lacking. However, our results suggest that HOTAIRM1, GAS5, MZF1-AS1, and OIP5-AS1 could be potential biomarkers to help clinicians accurately stratify different asthma phenotypes, which would have an indirect impact on the early diagnosis and quality of life of asthmatic patients.

Furthermore, our results suggest that OIP5-AS1 and lncRNA GAS5 could be a cornerstone that helps to explain how obesity is directly related to asthma. There is a significant knowledge gap where more molecular biomarkers are needed; we think that OIP-5AS1 and lncRNA GAS5 could help in the future to understand better molecular mechanisms involved in asthma and obesity pathophysiology [11]. In this line, Zsófia Gál and collaborators found OIP5-AS1 and MZF1-AS1 differentially expressed in whole blood of adults with allergic rhinitis vs. adults with chronic obstructive pulmonary disease, and higher expression levels of OIP5-AS1 from whole blood of adults with allergic asthma versus adults with non-allergic asthma [24].

Finally, the bioinformatics analysis showed us an exciting signature of genes and miRNAs. In this sense, since the discovery of mathematics, the human being has tried to find an explanation through numbers for our entire environment, and today thanks to the advancement of technology and, in this case of computing, it has become much easier to try to predict the behavior of biological systems with the help of diverse bioinformatics software and algorithms [32]. In this context, with the help of powerful bioinformatic tools, several studies have tried to predict different molecular pathways that are directly related lncRNAs, miRNAs, and genes, which have increased the knowledge about interactions between that kind of molecules [33,34]. It is worth mentioning that several databases have been created to systematize and visualize causal relationships between asthma and lung disease [35,36]. Although mathematics and computer science have helped generate much new knowledge in biological sciences, there is still a long way to go, and indeed in the future, much more powerful software and algorithms will be created to help advance disease research.

As a limitation, the design of our study is cross-sectional. Therefore no prospective results were explored. Thus, further experiments with the OIP5/GAS5 signature should be designed to test its potential ability to predict the development of asthma in the context of obesity-related asthma. However, this study raises an essential issue about the characterization of endotypes and varying phenotypes in asthma, which is a very interesting finding and could help future research development.

## 4. Materials and Methods

### 4.1. Study Design

A cross-sectional study was carried out from August 2019 to January 2020. All patients were sampled by convenience in the Children’s Hospital of Mexico Federico Gómez, Mexico City.

### 4.2. Subjects

To accomplish our objective and minimize the measurement bias, we performed the long noncoding RNAs quantification in 4 different groups: (1) healthy adolescents (HA), (2) allergic asthma without obesity (AA), (3) obesity without asthma (OB), and (4) obesity-related asthma (OA). Furthermore, according to the declaration of Helsinki (2013) and the Mexican Regulation for Biomedical Research, all participants’ parents signed informed consent, and adolescents firmed assent before any procedure. Therefore, the study was approved by the research, ethics in research, and biosecurity committee of the Children’s Hospital of Mexico Federico Gómez with the approval numbers HIM/2013/015.SSa.1601 and HIM/2020/030.SSa.1667.

### 4.3. RNA Extraction, cDNA Synthesis, and Real-Time PCR

RNA was isolated from 300 µL of peripheral blood, according to the Guanidine Thiocinate method (Trizol, Invitrogen, Waltham, MA, USA) [37]. The RNA obtained from each sample was quantified in a nanophotometer (IMPLEN). The 260/280 nm absorbance of each RNA sample was 1.9 + 0.2, consistent with the absence of contamination. The RNA integrity was evaluated in 1.5% agarose gels stained with Ethidium Bromide.

We used primers designed by QIAGEN to determine the expression of target genes: RT² lncRNA qPCR Assay for Human GAS5: (NCBI 60674), Catalog No.—#LPH-11340A-200_3410131; OIP5-AS1 (NCB1 729082), Catalog No.—#LPH15948A-200_3316050; MZF1-AS1 (NCBI 100131691), Catalog No.—# LPH25874A-200; HOTAIRM1 (NCBI 100506311), #LPH10483A-200_3442487 (Qiagen, Hilden, Germany). The reverse transcriptase (Improm II™, Promega, Madison, WI, USA) and the PCR (FastStart™ SYBR^®^ Green Master, Roche, Basel, Switzerland) were performed according to manufacturer instructions. Relative quantification was carried out by the 2^−∆∆Ct^ method using Glyceraldehyde-3-Phosphate Dehydrogenase (GAPDH) as a reference gene.

### 4.4. Statistical Analysis

Statistical analyses were performed using Statistical Package for Social Sciences v24.0 software (SPSS Inc., Chicago, IL, USA). To test the distribution of quantitative variables Kolmogorov-Smirnov test was performed. Quantitative data are presented as the medians and interquartile range (IQR). The Mann-Whitney U test was performed to analyze statistical differences between 2 groups, and the Kruskal-Wallis test was carried out to analyze statistical differences between more significant than two groups with Dunnet post hoc analysis. Fisher’s tests analyzed comparisons of proportions. Correlations were analyzed with a nonparametric Spearman correlation test. To evaluate the accuracy of predicting non-allergic asthma, receiver operating characteristic (ROC) curve analysis was performed, and the area under the curve (AUC) with the Youden’s index optimal cut-off point was calculated. A *p* < 0.05 with a 95% confidence interval (CI) was considered statistically significant.

### 4.5. Bioinformatic Analysis

An interaction network was created in which we searched the base-pairing interactions of studied lncRNAs with diverse miRNAs and Genes in diverse databases, such as RNA interactome Database (https://www.rna-society.org/rnainter/ (accessed on 25 August 2022)), RNA central (https://rnacentral.org/; https://diana.e-ce.uth.gr/lncbasev3 (accessed on 25 August 2022)), LNCipedia (https://lncipedia.org/ (accessed on 27 August 2022)) and LncRRIsearch (http://rtools.cbrc.jp/LncRRIsearch/ (accessed on 2 September 2022)), we represented the direct and indirect interactions that OIP5-AS1, MZF1-AS1, HOTAIRM1, and GAS5 can have with several genes that have previously described as critical components of the pathophysiology of both major endotypes of asthma.

Network analysis was performed with the mirNET 2.0 bioinformatic tool: https://www.mirnet.ca/ (accessed on 5 September 2022) [38]. And visualization was achieved through Cytoscape v3.9.1 software.

## Figures and Tables

**Figure 1 ijms-24-06481-f001:**
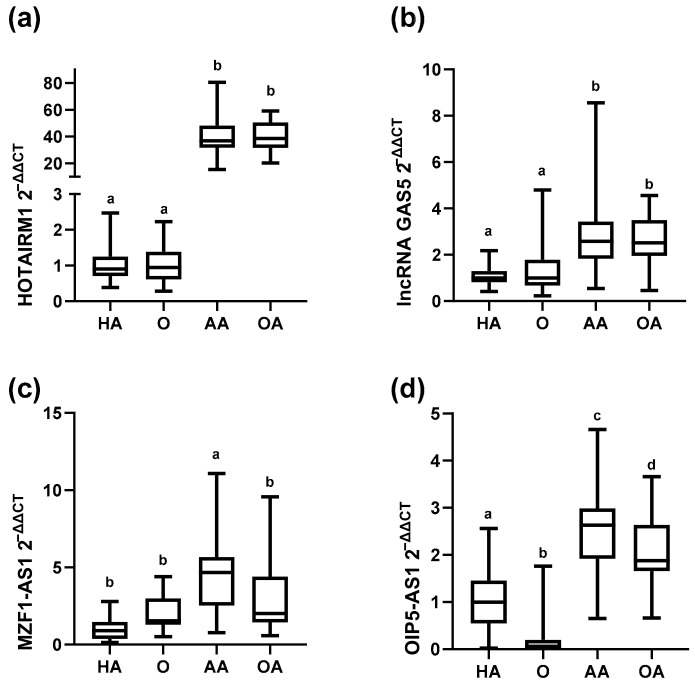
Differential expression of long noncoding RNAs: HOTAIRM1, GAS5, MZF1-AS1, and OIP5-AS1 from the four studied groups’ whole blood. (**a**) HOTAIRM1 expression levels comparison between studied groups. (**b**) lncRNA GAS5 expression levels comparison between studied groups. (**c**) MZF1-AS1 expression levels comparison between studied groups (**d**) OIP5-AS1 expression levels comparison between studied groups. The Kruskal-Wallis test was performed. Data are represented with Box and Whiskers plots. Clusters with the same letter code are not significantly different (Dunnett’s multiple comparison tests, *p* < 0.05, were considered statistically significant).

**Figure 2 ijms-24-06481-f002:**
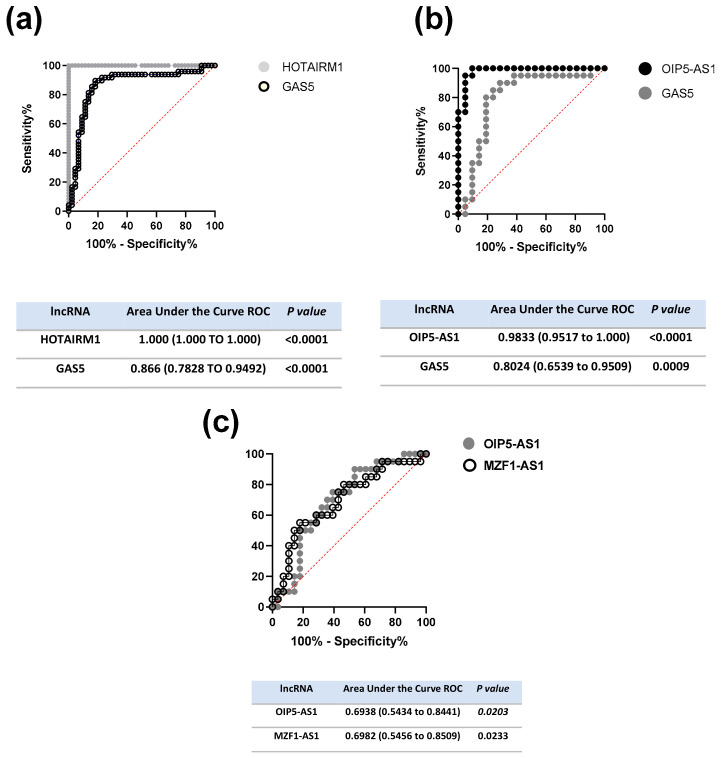
Discriminative value of lncRNAs between interest groups (**a**) Area under the curve ROC analysis of HOTAIRM1 and GAS5 to discriminate between two endotypes of asthma (AA and OA) from their respective controls (HA and OB groups); (**b**) Area under the curve ROC analysis of OIP5-AS1 and GAS5 to discriminate between obesity alone from obesity-related asthma patients; (**c**) Area under the curve ROC analysis of OIP5-AS1 and MZF1-AS1 to discriminate between two major asthma endotypes.

**Figure 3 ijms-24-06481-f003:**
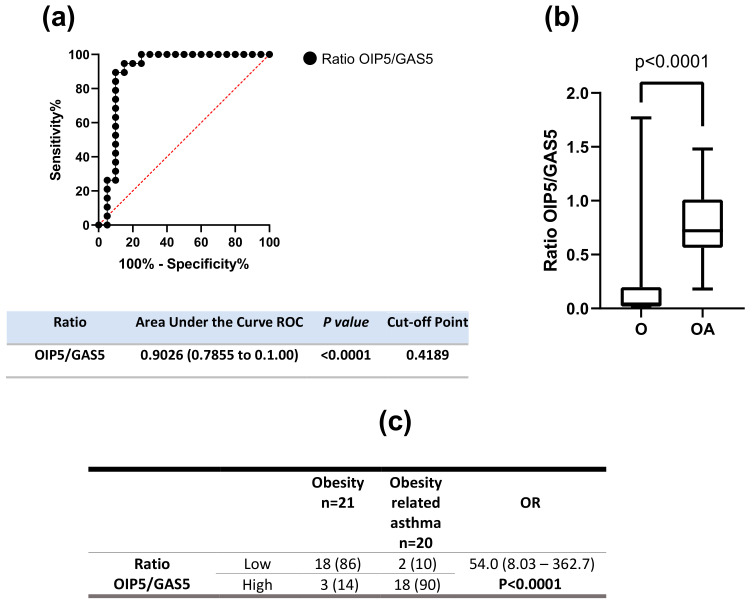
OIP5/GAS5 expression is associated with obesity-related asthma. (**a**) Area under the curve ROC analysis of OIP5-AS1/GAS5 ratio to discriminate between obesity alone from obesity-related asthma patients; (**b**) Mann-Whitney U test comparing OIP5/GAS5 ratio between obesity alone and obesity-related asthma groups; (**c**) Fisher’s Test Analysis with Odd’s ratio.

**Figure 4 ijms-24-06481-f004:**
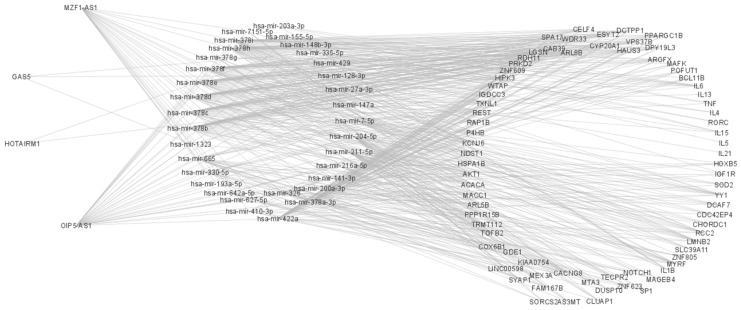
Interaction network of the lncRNAs studied in a lncRNAs-miRNAs-Genes network that could interact through base-pairing interaction. The analysis of networks was performed using the miRNET 2.0 bioinformatics tool and visualization was achieved through Cytoscape v3.9.1.

**Table 1 ijms-24-06481-t001:** Patient Characterization.

	HA (*n* = 23)	AA (*n* = 28)	OB (*n* = 21)	OA (*n* = 20)	*p*-Value
Age, years	16.0 (13.0–18.0)	13.0 (12.0–17.0)	14.0 (13.0–18.0)	14.0 (11.0–16.5)	0.1
Female, *n* (%)	12.0 (52.2)	17.0 (50.7)	14.0 (66.7)	9.0 (45.0)	0.5 *
Tanner, *n* II/III/IV/V	1/6/6/10	3/10/7/8	3/4/8/6	4/6/5/5	0.7 *
Weight, kg	54.0 (45.0–58.5)	47.9 (41.4–57.9)	74.8 (69.1–84.8) &#	71.4 (65.0–79.8) &#	<0.001
Height, cm	160.0 (154.0–165.0)	154.5 (146.8–164.8)	159.2 (155.5–161.5)	159.1 (153.0–165.0)	0.6
BMI, kg/m^2^	21.0 (17.8–23.4)	20.2 (18.5–21.1)	30.0 (27.4–32.4) &#	27.9 (26.8–31.1) &#	<0.001
BMI percentile	52.0 (43.0–65.0)	70.5 (39.5–76.8)	97.0 (96.0–97.5) &#	97.0 (96.0–98.1) &#	<0.001
BMI z-score	0.2 (0.0–0.3)	0.4 (−0.3–0.8)	1.8 (1.7–1.8) &#	1.8 (1.7–2.2) &#	<0.001
C. abdomen, cm	76.0 (70.4–83.0)	74.5 (70.2–80.0)	98.0 (92.5–100.2) &#	98.0 (92.1–100.6) &#	<0.001
C. waist, cm	74.2 (67.0–81.8)	72.5 (66.1–79.0)	92.4 (89.1–98.0) &#	90.5 (85.7–96.2) &#	<0.001
C. hip, cm	85.5 (78.3–90.8)	82.5 (79.2–89.8)	103.0 (98.0–110.8) &#	101.0 (94.3–107.9) &#	<0.001
Time with obesity, years			5.0 (4.5–5.0)	9.0 (7.3–10.8)	<0.001 ¶
Time with asthma, years		6.0 (5.0–7.5)		5.0 (5.0–6.0)	0.3 ¶
Severity of asthma					0.04 *
Mild intermittent *n*, (%)		6 (21.4)		1 (5.0)	
Mild persistent *n*, (%)		18 (64.3)		10 (50.0)	
Moderate persistent *n*, (%)		4 (14.3)		9 (45.0)	
Inhaled steroid *n*, (%) $		15 (53.6)		12 (60.0)	0.6 *
Antileukotriene *n*, (%)		5 (17.9)		7 (35.0)	0.2 *

* Pearson’s chi-square and ¶ Mann-Whitney U tests. Values are expressed as medians and interquartile ranges. Abbreviations: BMI, body mass index; C, circumference; HA, Healthy adolescents; AA, Allergic asthma without obesity; OB, Obesity without asthma; OA, Obesity-related asthma. $ Dosage is equivalent to 200–400 μg of budesonide. & *p* < 0.001 vs. healthy; # *p* < 0.001 vs. allergic asthma. The first column regarding HA is reproduced with permission from Elsevier, reference Leija-Martínez et al. [4].

## Data Availability

Raw data are available directly from the corresponding authors with a reasonable request.

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
