# Peer review of "Whole Blood Expression Levels of Long Noncoding RNAs: HOTAIRM1, GAS5, MZF1-AS1, and OIP5-AS1 as Biomarkers in Adolescents with Obesity-Related Asthma"

_ijms, 2023, doi:10.3390/ijms24076481_

Round 1
Reviewer 1 Report
The assignment of therapeutic treatment of asthma is challenging due to the different response in different patients. Accurate classification and discrimination of the different endotypes is still missing and the role of lncRNAs as potential biomarkers to discriminate between allergic asthma and obesity-related asthma in adolescents is still unknown.
In this manuscript, the authors focus on 4 lncRNAs, known to be involved in inflammation, obesity and lung diseases, to understand if they can be used as potential molecular biomarkers to distinguish the two main asthma endotypes, allergic asthma and obesity-related asthma, in adolescent population.
Thus, the authors use whole blood from healthy, obese, allergic asthmatic and obesity-related asthma adolescents to evaluate by qRT-PCR and different statistical analyses the levels of the different lncRNAs in the different groups and, the final conclusion is that these lncRNAs could help discriminate the obesity-related asthma.
Moreover, combining this data with the generation of interaction networks of diverse miRNAs and genes, the authors showed that all lncRNAs could be associate with critical genes involved in asthma pathophysiological pathways.
This study contributes to the field of lung pathologies, specifically in asthma, indicating potential lncRNAs to be used as biomarkers.
Author Response
Thank you for your valuable time and observations, we agree that this is a very interesting topic.
Reviewer 2 Report
The article is deal with the whole blood expression levels of long non-coding RNAs: HOTAIRM1, GAS5, MZF1-AS1 and OIP-5-AS1 as biomarkers in adolescents with obesity-related asthma. The topic discussed is very important for the treatment and prevention of asthma.
I would like to make a few comments:
1) Line 33:
“Abstract: Distinct endotypes and varying phenotypes; characterized by common….”
Add, please, the word “asthma”:
Abstract: Distinct asthma endotypes and varying phenotypes; characterized by common…
2) Line 293:
It is necessary to mention that various databases are being created to systematize and visualize causal relationships in asthma and lung disease, and cite the articles, please:
Boue S, Fields B, Hoeng J, Park J, Peitsch MC, Schlage WK, et al. Enhancement of COPD biological networks using a web-based collaboration interface. F1000Res. 2015 Jan 29;4:32. doi: 10.12688/f1000research.5984.2. PMID: 25767696; PMCID: PMC4350443.
Namasivayam AA, Morales AF, Lacave ÁM, Tallam A, Simovic B, Alfaro DG, et al. Community-Reviewed Biological Network Models for Toxicology and Drug Discovery Applications. Gene Regul Syst Bio. 2016 Jul 12;10:51-66. doi: 10.4137/GRSB.S39076. PMID: 27429547; PMCID: PMC4944831.
Author Response
1) Line 33:
We just added a few words that improve the line 33 that corresponds to abstract section, you will find it highlighted in color yellow in the new file.
2) Line 293:
We added a phrase and the suggested references mentioning that different databases have been created to systematize and visualize causal relationships in asthma and lung disease. (You will find it highlighted in color yellow in lines 266 and 267 in the attached file)
